# Cytological, Biochemical, and Transcriptomic Analyses of a Novel Yellow Leaf Variation in a *Paphiopedilum* (Orchidaceae) SCBG COP15

**DOI:** 10.3390/genes13010071

**Published:** 2021-12-28

**Authors:** Ji Li, Kunlin Wu, Lin Li, Meina Wang, Lin Fang, Songjun Zeng

**Affiliations:** 1Key Laboratory of South China Agricultural Plant Molecular Analysis and Gene Improvement, South China Botanical Garden, Chinese Academy of Sciences, Guangzhou 510650, China; liji17@scbg.ac.cn (J.L.); wkl8@scbg.ac.cn (K.W.); lilin@scbg.ac.cn (L.L.); 2University of Chinese Academy of Sciences, Beijing 100049, China; 3Shenzhen Key Laboratory for Orchid Conservation and Utilization/National Orchid Conservation Center of China/Orchid Conservation & Research Center of Shenzhen, Shenzhen 518114, China; snow-wmn2005@163.com; 4Guangdong Provincial Key Laboratory of Applied Botany, South China Botanical Garden, Chinese Academy of Sciences, Guangzhou 510650, China

**Keywords:** *Paphiopedilum* hybrid, leaf variegation, chlorophyll metabolism, transcriptome analysis

## Abstract

The genus *Paphiopedilum*, belonging to the Orchidaceae, has high ornamental value. Leaf variations can considerably improve the economic and horticultural value of the orchids. In the study, a yellow leaf mutant of a *Paphiopedilum* hybrid named *P.* SCBG COP15 was identified during the in vitro plant culture process; however, little is known about their molecular mechanisms. For this, RNA-seq libraries were created and used for the transcriptomic profiling of *P.* SCBG COP15 and the yellow mutant. The Chl *a*, Chl *b*, and carotenoid contents in the yellow leaves decreased by approximately 75.99%, 76.92%, and 56.83%, respectively, relative to the green leaves. Decreased chloroplasts per cell and abnormal chloroplast ultrastructure were observed by electron microscopic investigation in yellowing leaves; photosynthetic characteristics and Chl fluorescence parameters were also decreased in the mutant. Altogether, 34,492 unigenes were annotated by BLASTX; 1,835 DEGs were identified, consisting of 697 upregulated and 1138 downregulated DEGs. *HEMA*, *CRD*, *CAO*, and *CHLE*, involved in Chl biosynthesis, were predicted to be key genes responsible for leaf yellow coloration. Our findings provide an essential genetic resource for understanding the molecular mechanism of leaf color variation and breeding new varieties of *Paphiopedilum* with increased horticultural value.

## 1. Introduction

Leaves are vital organs in plants for photosynthesis, respiration, and nutrient transformation, as well as an important aesthetic trait in ornamental plants. Commonly, chlorophyll (Chl), carotenoids, and anthocyanins are the main leaf pigments that are determined by genotype and growth environment [1]. Chl is the primary photosynthetic pigment found in leaves. It is able to capture light energy and transfer it to the photoreaction center to generate chemical energy during photosynthesis [2]. Recently, most research has focused on red leaves, which are associated with anthocyanin accumulation [3]. However, the yellowing of leaves has received significantly less attention, and most studies on leaf yellowing have focused on Chl metabolism [4,5,6].

In *Arabidopsis thaliana*, a total of 27 genes participating in Chl biosynthesis have been identified (Appendix A), including those contributing to Glu-tRNA, Chl *a,* and Chl *b* production [2], and mutations at any position may lead to low Chl content, resulting in abnormal leaf coloration [7]. For instance, the RNA silencing of *HEMA* in barley caused varying degrees of inhibition of Chl biosynthesis, and as a result the plant showed an albino and yellow phenotype [8]. In rice, mutations in the *CHLI* and *CHLD* genes lead to a decrease in enzyme activity and Chl content [9]. Chl degradation is also one of the main reasons for leaf discoloration. Generally, the biosynthesis and degradation of Chl are in dynamic equilibrium within the metabolic process, which is essential for green plants; once the Chl degradation pathway related genes are abnormal, the plant will show the corresponding abnormal leaf color [10]. In rice, NYC1 and NOL can form a complex and act as a Chl *b* reductase to catalyze Chl degradation [11]. The high expression of *NYE1* in *A**. thaliana* causes yellowing and even albino leaves, and it was confirmed that NYE1 intervenes with the regulation of PAO activity during the Chl degradation process [12].

*Paphiopedilum*, also known as the Venus slipper orchid and Cinderella, has a high ornamental value due to its unique flower type, gorgeous flower color, long-lasting flowering period, and elegant, upright leaves. It has long become a very popular upscale flower in the international flower market [13]. In orchids, leaf variations such as striped leaves, yellow leaves, and spotted leaves have recently gained increased popularity among breeders and customers. Accordingly, these extraordinary traits significantly improve the economic and horticultural value of the plants [14,15,16,17,18]. Presently, large-scale production of *Paphiopedilum* seedlings is mainly performed through tissue culture [19]. A *Paphiopedilum* hybrid variety named *P.* SCBG COP15 has a small probability of developing golden leaves during its tissue culture process; however, the understanding of the underlying molecular mechanisms in the yellow leaf mutation remains limited. This phenotype is stably inherited; therefore, we obtained this yellow leaf mutant of *P.* SCBG COP15 for further investigation.

In this research paper, photosynthetic pigments and chloroplast ultrastructure, together with transcriptomics, were compared between yellow mutant leaves and the green normal leaves of *P.* SCBG COP15. We identified differentially expressed genes (DEGs) and transcription factors (TFs) related to Chl metabolism, and evaluated the expression levels of some key unigenes by quantitative real-time polymerase chain reaction (qRT-PCR) to validate their involvement in leaf variation. Our findings illustrate the physiological, cytological, and bioinformatic aspects of yellow leaf mutants of *P.* SCBG COP15, and they lay the foundation for further understanding of yellow leaf coloration and the facilitation of molecular breeding in *Paphiopedilum*.

## 2. Materials and Methods

### 2.1. Plant Material and Sample Preparation

One-year-old seedlings of the *Paphiopedilum* hybrid named *P.* SCBG COP15, along with the yellow leaf variety which formed through the tissue culture by using lateral bud explants were used in present study. *P.* SCBG COP15 was a combination between *P.* Maudiae as female parent and *P.* SCBG Yunzhijun as male parent, The experimental materials were grown in the tissue culture laboratory and greenhouse of the South China Botanical Garden, Chinese Academy of Sciences, Guangzhou, China (Appendix A). Fresh and mature leaves were collected for physiological observation, determination of photosynthetic characteristics, and detection of Chl fluorescence parameters. The samples were stored at −80 °C for biochemical experiments and RNA extraction. 

### 2.2. Pigment Content Measurement

Approximately 0.1 g fresh leaf tissue was extracted with 80% acetone overnight at 4 °C, and the absorbance was measured at 646.8 nm, 663.2 nm, and 470 nm in a 96-well plate with a microplate reader (Tecan Infinite, Männedorf, Switzerland). The Chl and carotenoid content was calculated according to Lichtenthaler [20].

### 2.3. Photosynthetic Characteristics and Chlorophyll Fluorescence Parameter Measurement

An LI-6800 portable photosynthesis system (Li-cor, Lincoln, NE, USA) was used to measure photosynthetic characteristic parameters, including intercellular CO_2_ concentration (Ci), stomatal conductance (Gs), and net photosynthetic rate (Pn). The fluorescence parameters were measured using a Chl fluorescence spectrometer (Heinz Walz GmbH, Effeltrich, Germany), with the leaves left in the dark for approximately 30 min. Finally, F_o_, F_m_, NPQ, qN, and Y(II), along with F_v_/F_m,_ were measured.

### 2.4. Anatomy Observations

Leaf samples of the normal wild type of *P.* SCBG COP15 (G), and its yellow mutant (Y), were cut into approximately 1 mm × 2 mm pieces and were fixed with 0.1 M phosphate buffer (pH = 7.2), containing 2.5% glutaraldehyde and 2% paraformaldehyde. After washing the leaf samples six times with the 0.1 M phosphate buffer, the samples were postfixed in 1% osmium tetroxide for 4 h and subsequently washed again with 0.1 M phosphate buffer. Ultrathin sections (80 nm) were cut using an ultramicrotome (Leica UC7, Leica Microsystems, Wetzlar, Germany), which were then stained with 4% uranyl acetate and 2% lead citrate. Ultrathin sections of the leaf ultrastructures were observed using transmission electron microscopy (TEM) (transmission electron microscope JEOL JEM-1010, Tokyo, Japan), operating at 100 kV. For the characterization of the leaf architecture, fresh leaf samples from G and Y were first fixed in 4% agar, and then cut into 200 µm tissue slices using a vibratome (Leica VT 1200, Leica Microsystems, Wetzlar, Germany) at 1.00 mm/s speed and 3.00 mm amplitude. The slices were observed under a Leica DVM6 ultra-depth field microscope (Leica Microsystems, Wetzlar, Germany). 

### 2.5. RNA Extraction, Library Construction, Sequencing

The leaf samples were ground into a fine powder in liquid nitrogen. Total RNA was isolated using an RNA kit (Polysaccharides & Polyphenolics-rich) (Hua Yueyang, Beijing, China), with the use of RNase-free DNase I (Takara Bio, Shiga, Japan) to remove genomic DNA contamination. The integrity and purity of the total RNA was determined using a 2100 Bioanalyzer (Agilent Technologies, Inc., Santa Clara, CA, USA) and quantified using the ND-2000 NanoDrop (Thermo Scientific, Wilmington, DE, USA). Only high-quality RNA samples were used to prepare RNA-seq libraries. Independent quadruplication of whole-leaf materials from *P.* SCBG COP15 and its yellow leaf variety were used to construct the libraries and sequences with a Illumina TruSeqTM RNA sample preparation Kit (San Diego, CA, USA) and the Illumina HiSeq™ 6000 sequencing platform (Illumina, San Diego, CA, USA) by Shanghai Majorbio Bio-pharm Biotechnology Co., Ltd. (Shanghai, China), respectively.

### 2.6. Transcriptome Assembly and Annotation

The total clean reads from the eight libraries were assembled de novo using Trinity software (http://trinityrnaseq.sourceforge.net/ 10 November 2021), with the use of the default parameters. The assembled unigenes were searched against the public databases, including the NCBI non-redundant protein (NR) database (http://www.ncbi.nlm.nih.gov), Swiss-Prot (http://www.expasy.ch/sprot/), Clusters of Orthologous Groups (COG) database (http://www.ncbi.nlm.nih.gov/COG), Gene Ontology (GO) functional annotation (http://www.blast2go.com/b2ghome) [21], and the Kyoto Encyclopedia of Genes and Genomes (KEGG) database [22].

### 2.7. DEGs Identification and Functional Enrichment Analysis

DEGs were identified using DESeq2 [23] and unigenes. A fold change (FC) >2 or <−2 and an adjusted *p*-value ≤ 0.05, considered the transcripts per million reads (TPM) significantly differentially expressed. All DEGs were then mapped to GO terms and KEGG pathways in the respective databases using Goatools (https://github.com/tanghaibao/ Goatools) and KOBAS (http://kobas.cbi.pku.edu.cn/home.do) [24].

### 2.8. Validation of DEGs by qRT-PCR

A total of 1 μg of high quality total RNA (the same sample used in RNA-Seq) was reverse-transcribed to first-strand cDNA using TransScript^®^ One-Step gDNA Removal cDNA Synthesis SuperMix (Transgen, Beijing, China) to perform qRT-PCR. Twelve unigenes related to Chl biosynthesis and degradation were selected for further validation. Specific primers were designed with the Primer Premier software (version 5.0), and qRT-PCR was performed in a 384-well block with PerfectStart Green qPCR SuperMix (Transgen, Beijing, China) on a LightCycler 480II (Roche, Mannheim, Germany). The reaction conditions were set as follows: 95 °C for 30 s, 40 cycles at 94 °C for 5 s, and 60 °C for 30 s. Cq values were analyzed using the LightCycler^®^ 480 software. The relative unigene expression levels of target genes were determined by the 2^−ΔΔCT^ method [25], with the actin gene (TRINITY_DN13664_c0_g3) as the internal control. 

## 3. Results

### 3.1. Leaf Anatomical Characteristics and Ultrastructure

As shown in Figure 1D, Chl was present throughout the whole mesophyll cell, and no pigment was observed in the leaf epidermis in neither the green nor yellow leaves. The mutant leaves showed a light-yellow color compared to the green color in the normal leaves. Furthermore, we found that chloroplasts showed the typical structures of green leaf mesophyll cells, with normal grana and thylakoids present. However, only a few thylakoids remained in the chloroplasts of yellow mutants and showed damaged thylakoid membranes, absent stromal lamellae, and a large number of irregularly arranged round vesicles (Figure 2). Additionally, the average number of chloroplasts in each cell in normal green leaves was 1.89-fold higher than that in yellow mutant leaves (Figure 3B). This result indicates that the variation in the leaf color of the yellow mutant might be a consequence of damage from chloroplast development.

### 3.2. Photosynthetic Characteristics and Chlorophyll Fluorescence Parameters

Six Chl fluorescence parameters were ascertained in the present study, and it was obvious that all parameters of the normal green leaf were higher than those of the yellow mutant; specifically, the F_v_/F_m_ value of yellow leaves was only 41.7% of the green leaves’ value (Figure 4), indicating that light absorption and energy transfer of the light-harvesting complexes were more efficient in the normal green leaf than in the yellow mutant. Next, the photosynthetic characteristic parameters, including intercellular CO_2_ concentration (Ci), stomatal conductance (Gs), and net photosynthetic rate (Pn), were determined for further analysis. The Ci, Gs, and Pn values of the green leaves were approximately 35.93, 73.36, and 60.24% higher than those of the yellow leaf (Figure 4B–D), respectively, suggesting a stronger photosynthetic capacity in green leaves compared to the mutant.

### 3.3. Library Construction and De Novo Assembly

To reveal the molecular mechanism of the yellow-leaf phenotype in *P.* SCBG COP15, high-quality RNA samples extracted from the leaves of G and Y were applied to construct the sequencing library with four biological replicates (Appendix A). The number of clean reads for each library ranged from 40,802,964 to 53,349,426, with a mapped ratio of 67.95% to 71.91%. A total of 79,129 unigenes were identified from the mapped libraries, as well as 113,567 transcripts obtained, with an N50 length of 1590 bp. The GC content ranged from 47.77% to 49.43%, and the Q30 base percentage was ≥94.71%, which indicated a high read confidence level (Appendix A).

### 3.4. Functional Annotation and Classification

In total, 34,492 unigenes were annotated in the NR, Nt, SwissProt, KEGG, COG, Pfam, and GO databases using BLASTX, which accounted for 43.59% of the assembled unigenes. In the NR annotated species distribution, the top three matched species were *Dendrobium catenatum*, *Phalaenopsis equestris*, and *Apostasia shenzhenica* (Appendix A). Of these annotated unigenes, 28,116 (35.53%) were categorized as contributing to a biological process, cellular component, or molecular function in the GO database. Binding, catalytic activity, and cellular processes were the three prominent subclasses. A total of 9364 unigenes were annotated in the KEGG functional classification, and unigenes accounted for the highest percentage of translation, carbohydrate biosynthesis and folding, sorting, and degradation (Appendix A).

### 3.5. DEGs Analysis and Verification 

DEGs of Y and G were analyzed based on the TPM values of the unigenes. A total of 1835 DEGs were identified (P adj. ≤ 0.05, FC ≥ 2) by pairwise comparison, with 697 upregulated and 1138 downregulated DEGs observed. To obtain a functional categorization of the DEGs, they were annotated for GO analysis. Next, 1104 DEGs were categorized in three main GO classification categories, among which polysaccharide metabolic process, polysaccharide catabolic process, and extracellular region were the three most frequently identified terms (Figure 5). In addition, KEGG annotation was performed to identify the specific biochemical pathways involved in leaf variegation, with 463 DEGs successfully assigned to 111 KEGG pathways. The top 20 enriched KEGG pathways are displayed in Figure 6, and the most frequently represented pathways were plant hormone signal transduction, phenylpropanoid biosynthesis, and starch and sucrose metabolism, with 29, 21, and 17 DEGs observed, indicating that plant hormones appear to be playing an important role in the formation of plants leaf yellowing.

### 3.6. Expression Pattern of the Genes Involved in Chlorophyll Biosynthesis and Degradation

In the present study, 15 unigenes related to Chl biosynthesis and 11 unigenes associated with Chl degradation were identified based on RNA-seq annotation. *HEMA*(TRINITY_DN6358_c0_g1), the first structural gene in the Chl biosynthesis pathway, was significantly downregulated in the mutant, indicating that its decreased expression might contribute to the low Chl biosynthetic efficiency of Glutamate-1-semialdehyde (GSA). Furthermore, the protein and enzymatic chloroplastic precursor coding genes *CRD* (TRINITY_ DN2556_c0_ g1) and *CAO* (TRINITY_DN1555_c0_g1), respectively, also showed low expression in the yellow leaves. In contrast, the Chl degradation genes, especially *NYC1-1* (TRINITY_DN221_c0_g1) and *PPH3* (TRINITY_DN3203_c0_g1), showed significantly increased mRNA levels in the yellow leaves (Figure 7). This result indicates that the downregulation of key Chl biosynthesis genes, along with the upregulation of Chl degradation genes, may directly cause the decrease of Chl and consequently lead to the yellow leaf phenotype. We also identified that a unigene for the probable transcription factor GLK1 (ID: TRINITY_DN5163_c0_g1, *D**. catenatum*) was upregulated 2.11-fold in the normal green leaves compared with the mutant, which also indicates the important role of GLK (Golden 2-like) in leaf coloration. Moreover, the *GLK* gene family acts as a vital transcription factor to regulate chloroplast development, leaf color, hormone signal transduction, biological and abiotic stress and plant senescence, etc [26,27,28]. CDS and protein sequences that participated in Chl biosynthesis and degradation were screened from RNA-Seq data and cloned with the assistance of PCR to confirm the accuracy of the analysis and for further study (Appendix A).

### 3.7. TFs Involved in Leaf Coloration

TFs regulate the complex transcription network and therefore play a crucial role in regulating gene expression in a series of plant biological processes. In the present study, we identified 102 differentially expressed TFs belonging to 25 TF families, and the top five abundant TF families were the MYB superfamily (16, 15.6%), NAC (16, 15.6%), C2C2 (12, 11.7%), bHLH (10, 9.8%), and AP2/EPF (7, 6.8%). We analyzed 20 DEGs that exhibited highly significant downregulation or upregulation between yellow and green leaves and found that bHLH TFs were downregulated in the yellow mutant leaves, while most MYB superfamily TFs showed an upregulation trend in the mutant compared to normal green leaves. This suggests the TF’s vital function in leaf coloration (Figure 8; Appendix A).

### 3.8. Validation of Gene Expression Profiling

Twelve candidate unigenes involved in Chl biosynthesis and degradation were selected to test the validity of the transcriptomic data using qRT-PCR analysis. HEMA, CHLE and CAO participated in Chl biosynthesis, their genes showing a significant high expression level in green leaves, while Chl degradation genes including *NYC1-1*, *HCAR*, *PPH1*, *PPH2*, and *RCCR* were dramatically downregulated in green leaves. The results showed that the expression patterns revealed by qRT-PCR analysis were consistent with the transcriptomic data (Figure 9). Primers used in qRT-PCR was provided in Appendix A.

## 4. Discussion

*Paphiopedilum* is one of the most fashionable and rare orchid genera because of its high ornamental value, while wild populations of *Paphiopedilum* are facing the threat of extinction due to excessive collection and habitat destruction [29]. Presently, large-scale in vitro propagation is the main option for producing *Paphiopedilum* seedlings [30]. It is a common phenomenon that plant somatic cells undergo somaclonal variation during tissue culture. Therefore, this method is usually used to induce and obtain mutants in the process of plant breeding [31], which has been reported in wheat [32], the *Chrysanthemum* genus [33], *Saintpaulia* [34], and the *Cymbidium* genus [35]. In the present study, the yellow leaf mutant along with *P.* SCBG COP15, the green normal leaf hybrid discovered during the tissue culture process, were used as the materials to study leaf color variation in *Paphiopedilum*. Our results not only identified candidate structural and regulatory genes involved in leaf coloration, but also provided insight into golden leaf formation in *Paphiopedilum.*

Chl is the main pigment that harvests solar energy in leaf tissues and makes the leaves appear green. In the present study, the photosynthetic pigment content, including Chl *a*, Chl *b*, and carotenoid, in the green normal leaves was significantly higher than that of the mutant. Many Chl deficient mutants were reported in maize [36] and *A*. *thaliana* [37]. Generally, thylakoid membranes, arranged regularly and stacked into grana in the chloroplasts, are vital for chloroplast function, providing a platform for the photosynthetic protein pigment complexes and the conversion of energy throughout the process of photosynthesis [38,39]. We performed TEM in our study and discovered a strong contrast between the chloroplast ultrastructure of yellow and green leaves. The ultrastructure of mutant chloroplasts had serious defects, with some chloroplasts containing broken thylakoid membranes packed with vesicles and filled with a large number of plastid spheres. Furthermore, the chloroplast number in the mutant leaves decreased in comparison to wild type green leaves. In contrast, the structures in the chloroplasts of green leaves were intact, clearly visible, and well-organized. This result indicates that the yellow leaf phenotype might be a consequence of disordered chloroplast development. Similar results were also reported in the green leaves of *Ginkgo biloba* [6], rice [39], *Pseudosasa japonica* [40], *Ilex*
*× altaclerensis* [41] and *Anthurium andraeanum* [42].

Transcriptomic analysis has been widely used to identify key genes that are differentially expressed at different developmental stages, or under various physiological conditions. However, the genome of *Paphiopedilum* is still relatively unknown. In this study, RNA-sequencing analysis on *P.* SCBG COP15 leaf phenotypes (Y and G) at the same development stage were sequenced and annotated. Finally, we obtained 113,567 transcripts with N50 lengths of 1800 bp, and 79,129 unigenes with N50 lengths of 1590 bp, similar to those in other orchid species, such as *Cymbidium longibracteatum* [35] and *A**. shenzhenica* [43], respectively. Further, a total of 1835 DEGs were identified by TPM, with 463 DEGs successfully assigned to 111 KEGG pathways. The most frequently represented pathway was the plant hormone signal transduction. The results suggested that plant hormone is playing an important role in the formation of plants leaf yellowing, which has also been reported in the *Arabidopsis* genus [44,45,46].

Chl biosynthesis is catalyzed by 15 kinds of enzymes, and any disturbance in this process results in Chl deficiency and leaf color mutation [47]. *HEMA* is the first structural gene in Chl biosynthesis, encoding Glutamyl-tRNA reductase which catalyzes the initial substrate glutamyl-tRNA to glutamate-1-semialdehyde (GSA), with the unstable intermediate GSA then being isomerized to 5-aminolevulinic acid by GSA aminotransferase [48]. Previous research has shown that the mutation of the *HEMA* gene in rice causes the entire Chl synthesis pathway to be blocked, with the apparent yellow plant leaves visible [49], and RNA silencing of *HEMA* in barley showed a similar phenotype [8]. In *Arabidopsis*, *HEMA1* mutants are patchy to completely yellow and cannot grow healthily under normal growth conditions [50]. From our transcriptomic analysis, a *HEMA* (TRINITY_DN6358_c0_g1) gene was identified as a critical structural gene responsible for yellow mutant formation, due to its decreased expression level. In addition, *CRD* (TRINITY_ DN2556_c0_ g1), *CAO* (TRINITY_DN1555_c0_g1), and *CHLE* (TRINITY_DN2556_c0_g1) also showed low mRNA levels in yellow leaves. In contrast, the Chl degradation genes *NYC1-1* (TRINITY_DN221_c0_g1), *PPH* (TRINITY_DN16260_c0_g1 and TRINITY_DN3203_c0 _g1), and *HCAR* (TRINITY_DN482_c0_g1) had significantly increased expression levels in the yellow leaves, which indicates that the upregulated degradation genes in the mutant may accelerate Chl breakdown and lead to the yellowing of plant leaves. Specific expression levels of TPM related to Chl biosynthesis and degradation are provided in Appendix A.

## 5. Conclusions

We performed comparative analysis to investigate the differences in coloration between normal green leaves and yellow mutants of a *Paphiopedilum* hybrid. The low Chl content and abnormal ultrastructure of chloroplasts in the leaves of the yellow mutant suggested that Chl biosynthesis was partially inhibited, which could explain the yellow phenotype from both cytological and physiological aspects. Key structural genes related to Chl biosynthesis and degradation, along with potential transcription factors, were identified by DEG analysis from transcriptomic data. In summary, from our cytological, physiological, and transcriptomic analyses, decreasing the amount of photosynthetic pigments, blocking chloroplast development, and changing the expression levels of genes involved in Chl biosynthesis and degradation may collectively lead to the yellow leaf phenotype (Figure 10). Our findings provide an essential genetic resource, not only for the study of molecular mechanisms involved in leaf color variation, but also for the breeding of new varieties of *Paphiopedilum*.

## Figures and Tables

**Figure 1 genes-13-00071-f001:**
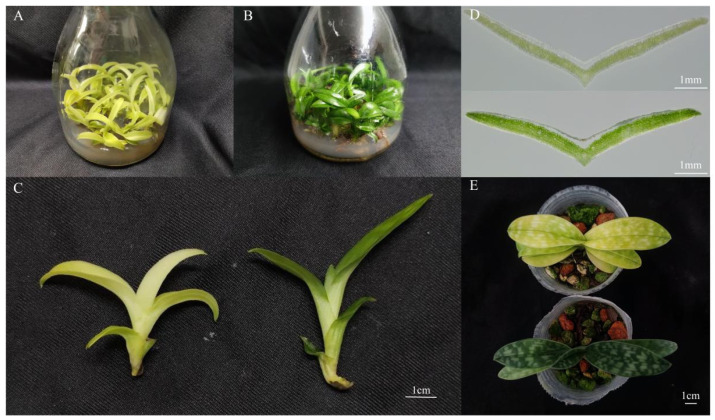
Phenotype of *P.* SCBG COP15 and the yellow mutant. (**A**) Phenotype of *P.* SCBG COP15. (**B**) Phenotype of the yellow mutant. (**C**) One year old seedling of *P.* SCBG COP15 (right) compared with the yellow leaf variety (left). (**D**) The anatomical distribution of pigments in *P.* SCBG COP15 (below) and the yellow mutant leaves (above). (**E**) Seedlings of the *P.* SCBG COP15 (below) and the yellow leaf variety (above) growing in the Orchid greenhouse.

**Figure 2 genes-13-00071-f002:**
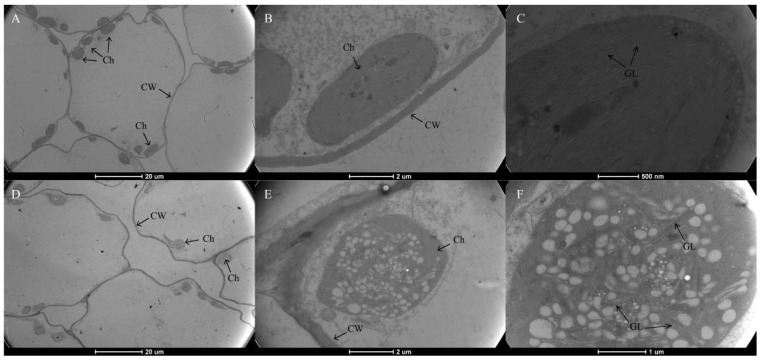
Transmission electron micrograph of chloroplasts from *P.* SCBG COP15 and the yellow mutant. (**A**–**C**) Chloroplast ultrastructures in the green leaves were intact, clearly visible, and well-organized. (**D**–**F**) The ultrastructure of mutant chloroplasts had serious defects, were packed with vesicles, and are filled with many plastid spheres. Ch: chloroplast; CW: Cell Wall; GL: grana lamella.

**Figure 3 genes-13-00071-f003:**
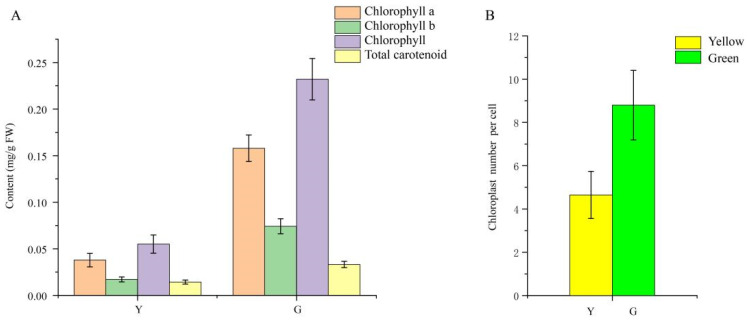
Pigment accumulation and the average number of chloroplasts per cell in *P.* SCBG COP15 and the yellow mutant. (**A**) Chlorophyll a, chlorophyll b, chlorophyll, and total carotenoid contents between Y (yellow mutant) and G (green normal leaves). (**B**) The average number of chloroplasts per cell decreased by 47.16% in the mutant leaves.

**Figure 4 genes-13-00071-f004:**
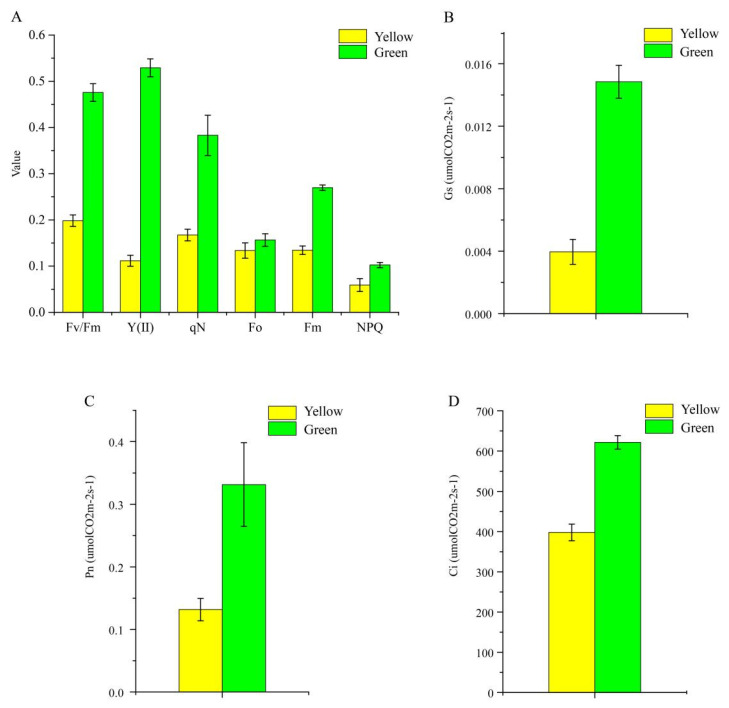
Photosynthetic characteristics (**A**) and three chlorophyll fluorescence parameters (**B**–**D**) of *P.* SCBG COP15 and the yellow mutant.

**Figure 5 genes-13-00071-f005:**
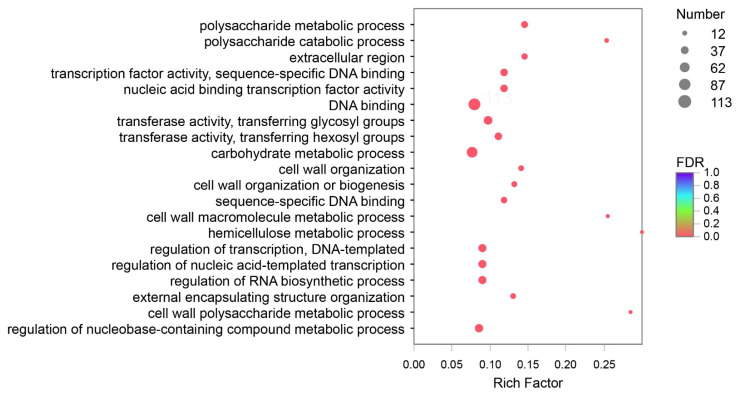
Gene Ontology (GO) enrichment analysis with the differentially expressed genes (DEGs). A total of 1104 DEGs were categorized in three main GO classification categories.

**Figure 6 genes-13-00071-f006:**
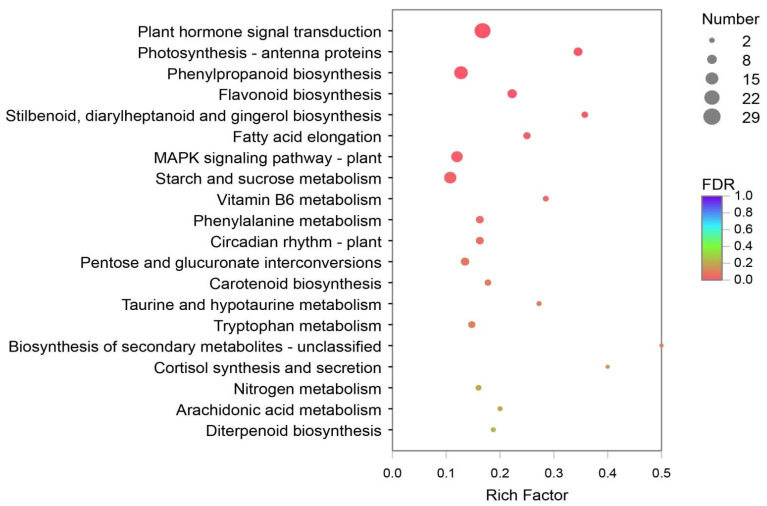
The Kyoto Encyclopedia of Genes and Genomes (KEGG) enrichment analysis with the DEGs. A total of 463 DEGs were assigned to 111 KEGG pathways, and the plant hormone signal transduction pathways were most frequently represented.

**Figure 7 genes-13-00071-f007:**
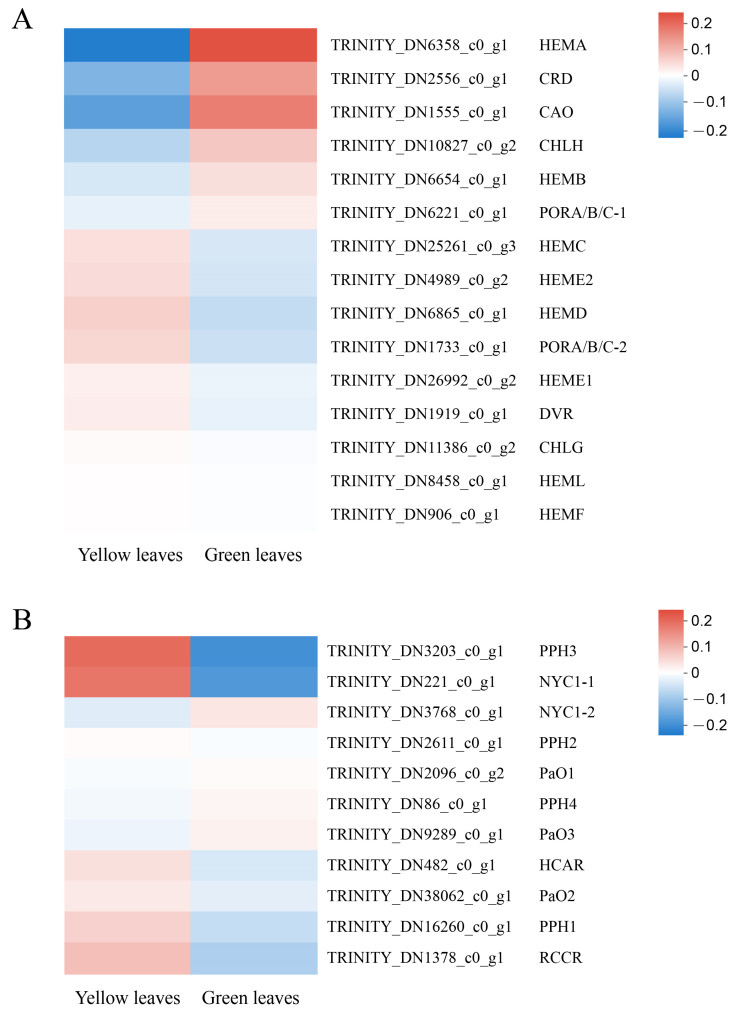
Expression profiles of unigenes involved in chlorophyll biosynthesis and degradation. The computing method of the expression profiles: The logarithm value of unigene TPM in each sample was first taken, and then hierarchical clustering is used for standardization. (**A**) 15 unigenes related to Chl biosynthesis. (**B**) 11 unigenes associated with Chl degradation.

**Figure 8 genes-13-00071-f008:**
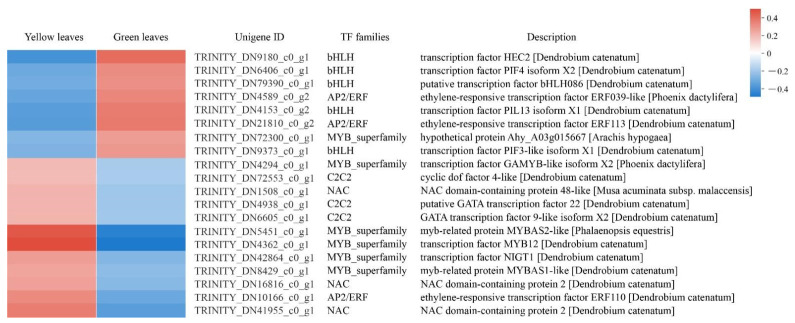
DEGs analysis of transcription factors (TFs) in normal green leaves and mutant leaves. Expression profiles and RNA-seq description of 20 DEGs which exhibited highly significant down or upregulation between yellow and green leaves.

**Figure 9 genes-13-00071-f009:**
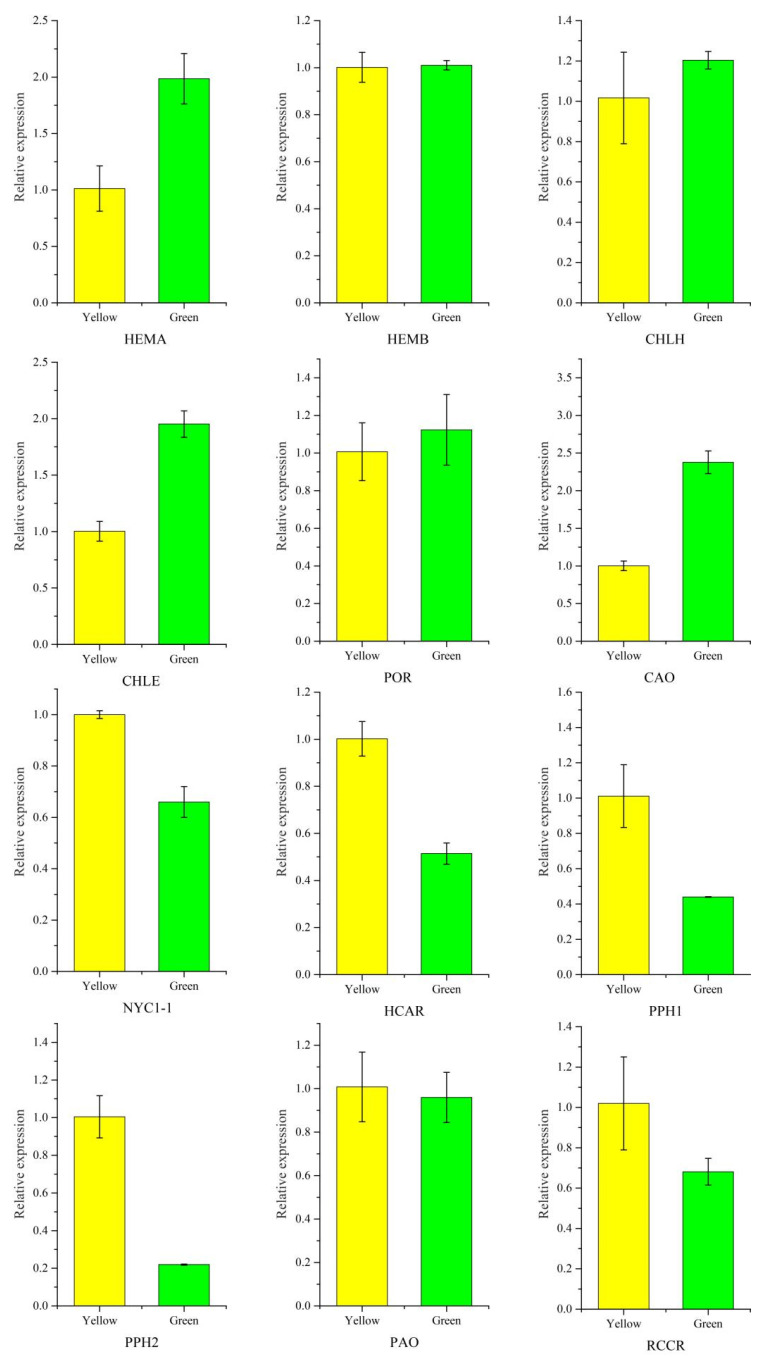
Validation of chlorophyll-related gene expression between *P.* SCBG COP15 and the yellow mutant. Expression levels were calculated by the 2^−ΔΔCT^ method with the actin gene (TRINITY_DN13664_c0_g3) as the internal control.

**Figure 10 genes-13-00071-f010:**
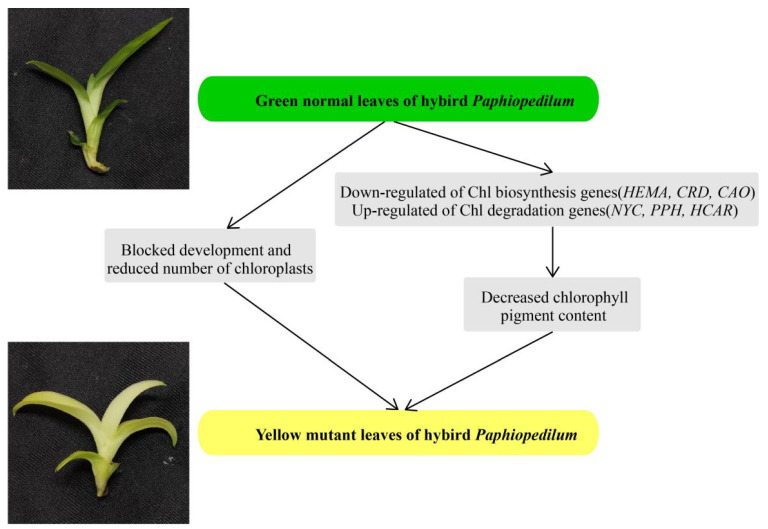
A proposed model for the mutant leaf coloration in the *Paphiopedilum* hybrid.

## Data Availability

Data may be found within the article or Appendix A. Raw data available upon reasonable request.

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
