# Peer review of "Cytological, Biochemical, and Transcriptomic Analyses of a Novel Yellow Leaf Variation in a Paphiopedilum (Orchidaceae) SCBG COP15"

_genes, 2021, doi:10.3390/genes13010071_

Round 1

Reviewer 1 Report

This article analyses the yellow color development in the Paphiopedilum orchid plants. The authors compared the green and yellow color variants for chlorophyll content and chlorophyll biosynthesis and degradation pathways. They also analyzed the genes related to the above pathways.

Overall these experiments are well designed, and article content is relevant for the journal.

comments

  1. Are there any carotenoid cleavage enzymes(CCD) reported in  Paphiopedilum plants? If present, did you check the expression of CCDs in both variants? Because this may contribute to the carotenoid accumulation in the yellow variant.

Author Response

Comments and Suggestions for Authors

This article analyses the yellow color development in the Paphiopedilum orchid plants. The authors compared the green and yellow color variants for chlorophyll content and chlorophyll biosynthesis and degradation pathways. They also analyzed the genes related to the above pathways.

Overall these experiments are well designed, and article content is relevant for the journal.

We thank the reviewer for this positive outlook on our manuscript.

Comments:

Are there any carotenoid cleavage enzymes(CCD) reported in Paphiopedilum plants? If present, did you check the expression of CCDs in both variants? Because this may contribute to the carotenoid accumulation in the yellow variant.

We carefully examined the transcriptome data and found one unigene (ID:TRINITY_DN4642_c0_g1) with description of putative carotenoid cleavage dioxygenase 4, chloroplastic [Apostasia shenzhenica], The average TPM value in the yellow leaves is 148, compared to 123 in the green leaves, The data showed no significant difference.

Reviewer 2 Report

I have carefully read MS which was submitted for consideration in the Genes (MDPI). The paper is in general well written, logically structured, well-illustrated and easy to understand. It also addresses a subject that is of great interest in the scientific community. The title clearly describes the contents of the paper. The abstract is well written. It encapsulates the entire study (a bit of introduction, aim, result and outcome). However, if there the section could be shortened a bit to make it more concise, it would be great. The introduction is well written as it gives a good background of the research in question. Also, the aim of the study is evident in the beginning and concluding parts. I believe that the Materials and Methods section is well structured and scientifically sound. The results are well presented, however, some figures need to be improved. Literature reviews in the discussion section of the manuscript are very good.

Generally, I have no critical comments, the work is very well prepared, I would recommend the publication of this manuscript.

Minor comments:

Title: Please consider supplementing the title with Latin names of plant family Orchidaceae, i.e. … in a Paphiopedilum (Orchidaceae) …, this can help search engines such as Google find your article and support its citations.

Line 66: Please delete the period in the sentence “…Yunzhijun). which…”

Line 256:  Please explain this sentence, maybe add something more.

Figure 7. This figure is unclear, please correct or prepare a new figure (please consider changing the graphic design, maybe just change the colours).

Line 329: Please add references in the first sentence: “Paphiopedilum is one of the most fashionable and rare orchid genera because of its  high ornamental value, while wild populations of Paphiopedilum are facing the threat of extinction due to excessive collection and habitat destruction [?].”

Line 331: The Latin phrase “in vitro”, please write in italics.

Line 358: Ginkgo biloba L. Please remove the abbreviation of the author of the name "L.", because other Latin names of taxa in the manuscript are written without abbreviations of the author's names.

Line 427: Latin orchid names:  Dendrobium catenatum, Phalaenopsis equestris, and Apostasia shenzhenica should be italicized.

Author Response

Comments and Suggestions for Authors

I have carefully read MS which was submitted for consideration in the Genes (MDPI). The paper is in general well written, logically structured, well-illustrated and easy to understand. It also addresses a subject that is of great interest in the scientific community. The title clearly describes the contents of the paper. The abstract is well written. It encapsulates the entire study (a bit of introduction, aim, result and outcome). However, if there the section could be shortened a bit to make it more concise, it would be great. The introduction is well written as it gives a good background of the research in question. Also, the aim of the study is evident in the beginning and concluding parts. I believe that the Materials and Methods section is well structured and scientifically sound. The results are well presented, however, some figures need to be improved. Literature reviews in the discussion section of the manuscript are very good.

Generally, I have no critical comments, the work is very well prepared, I would recommend the publication of this manuscript.

We thank the reviewer for this positive outlook on our manuscript.

Minor comments:

Title: Please consider supplementing the title with Latin names of plant family Orchidaceae, i.e. … in a Paphiopedilum (Orchidaceae) …, this can help search engines such as Google find your article and support its citations.

We accepted this suggestion and revised the title.

Line 66: Please delete the period in the sentence “…Yunzhijun). which…”

We accepted this suggestion and deleted this section.

Line 256:  Please explain this sentence, maybe add something more.

We accepted this suggestion and change the sentence as follow to make the context more clearer : The top 20 enriched KEGG pathways are displayed in Figure 6, and the most frequently represented pathways were plant hormone signal transduction, phenylpropanoid biosynthesis, and starch and sucrose metabolism, with 29, 21, and 17 DEGs observed, indicating that plant hormones appear to be playing an important role in the formation of plants leaf yellowing.

Figure 7. This figure is unclear, please correct or prepare a new figure (please consider changing the graphic design, maybe just change the colours).

We accepted this suggestion and add a detailed chart notes in the manuscript.

Line 329: Please add references in the first sentence: “Paphiopedilum is one of the most fashionable and rare orchid genera because of its  high ornamental value, while wild populations of Paphiopedilum are facing the threat of extinction due to excessive collection and habitat destruction [?].”

We accepted this suggestion and add a references in this sentence.

Line 331: The Latin phrase “in vitro”, please write in italics.

We accepted this suggestion and revised the manuscript.

Line 358: Ginkgo biloba L. Please remove the abbreviation of the author of the name "L.", because other Latin names of taxa in the manuscript are written without abbreviations of the author's names.

We accepted this suggestion and revised the manuscript.

Line 427: Latin orchid names:  Dendrobium catenatumPhalaenopsis equestris, and Apostasia shenzhenica should be italicized.

We accepted this suggestion and revised the manuscript.
